

# The usefulness of balance test in preseason evaluation of injuries in amputee football players: a pilot prospective observational study

Zofia Kasińska[1], Tomasz Tasiemski[1], Teresa Zwierko[2], Piotr Lesiakowski[3] and Monika Grygorowicz[4,5]

[1] Department of Adapted Physical Activity, Poznań University of Physical Education, Poznan, Poland
[2] Laboratory of Kinesiology in Functional and Structural Human Research Centre,
Department of Physical Culture and Health, Institute of Physical Culture Sciences, University of Szczecin, Szczecin, Poland
[3] Department of Physical Education and Sport, Pomeranian Medical University, Szczecin, Poland
[4] Department of Physiotherapy, Poznan University of Medical Sciences, Poznan, Poland
[5] Sports Science Research Group, Rehasport Clinic FIFA Medical Centre of Excellence, Poznan, Poland

Corresponding author
Monika Grygorowicz,
monika.grygorowicz@rehasport.pl,
grygorowicz@ump.edu.pl

## ABSTRACT

**Background**. Low balance ability is generally associated with an increased risk of ligament injuries. It seems that assessing the level of stability in amputee football players can help evaluate the accompanying risk of sports injuries. Thus, the study aimed to examine the usefulness of the balance test in preseason evaluation by calculating between-group differences in stability parameters between injured and non-injured amputee players.

**Methods**. The study was designed as a pilot prospective observational study. Twenty-five elite amputee football players representing the Polish National Team and the highest division in Polish League completed one-leg preseason balance tests on the Biodex Balance System before the start of the football season. All players in this study were male, with an average age of 29 years (SD = 7.9), a stature of 174.2 cm (SD 5.2) and a body mass of 80.1 kg (SD = 13.1). Then, players were prospectively observed over one football season, and lower leg injury data were prospectively collected through the nine months. The between-group differences were tested using the non-parametric Mann-Whitney $U$ test for players who sustained an injury (yes) and those who did not within the analysed season (no). Overall (OSI), medial-lateral (MLSI), and anterior-posterior stability index (APSI) were analysed as primary outcomes.

**Results**. The preseason values of the balance tests were not predictive ($p > 0.05$) regarding sustaining an injury during the season. No between-group differences were noted for any analysed outcomes ($p$ values ranged from 0.093 to 0.453).

**Conclusions**. Although static balance tasks offer a chance to make a preliminary assessment of injury prediction in amputee footballers, in overall, the balance test results cannot be regarded as the sole predictive injury risk factor in amputee football.

## INTRODUCTION

Balance is defined as the ability to maintain equilibrium and stability during various movements and activities, and it involves the intricate coordination of sensory information from multiple systems, including the visual, vestibular (inner ear), and somatosensory (proprioceptive) systems, which provide information about body position, motion, and orientation in space (*Twist, Gleeson & Eston, 2008*; *Reimann & Guskiewicz, 2000*; *Proske & Gandevia, 2012*). When regarding sports performance, it is particularly crucial for athletes (including amputee footballers), as it serves as a foundational skill that underpins various athletic movements, such as running, jumping, changing direction, and maintaining control while making precise movements, and it enhance an athlete's agility, speed, or accuracy (*Aytar et al., 2012*; *Jadczak et al., 2019*; *Hammami et al., 2021*).

Balancing is crucial for individuals with gait impairments, such as lower-limb amputation, as it is fundamental for executing complex motor skills and preventing falls and associated injuries (*Thomas et al., 2019*). Studies have linked increased postural sway and impaired postural adjustment processes to balance control issues in lower-limb amputees (*Ku, Abu Osman & Wan Abas, 2014*). While sports involvement can offer significant physical, mental, and social benefits to those with disabilities, it also presents a considerable injury risk (*Ristolainen et al., 2009*; *Sallis et al., 2001*; *Bahr & Holme, 2003*). Sports-related injuries and the fear of sustaining them can act as barriers to sports participation and impact an individual's life in numerous ways (*Laux et al., 2015*). Athletes with disabilities face challenges in accessing appropriate healthcare services in case of sports injuries, which can exacerbate their situation (*Vanlandewijck & Thompson, 2011*). The consequences of severe sports-related injuries on athletes with disabilities go beyond their athletic performance and directly influence their ability to perform daily living activities (*Bauerfeind et al., 2015*; *Fagher & Lexell, 2014*; *Ferrara & Peterson, 2000*; *Urbanski, Conners & Tasiemski, 2021*).

Several factors contribute to injuries among athletes with disabilities, including the type of disability or sports discipline, prosthetic adaptations, and skill level (*Willick & Webborn, 2011*). These factors intersect with sport demands and player attributes, impacting injury risk. While some researchers argue that the incidence of sports injuries is similar between able-bodied and disabled athletes, those involved in Paralympic sports typically experience minor injuries, allowing them to resume training in a short time (*Ferrara & Peterson, 2000*; *Nyland et al., 2000*). The injury rate among athletes with disabilities stands at 9.3 injuries per 1000 athlete-exposures, lower than American football and soccer but higher than basketball (*Ferrara & Peterson, 2000*). Understanding the demands of sports and player characteristics is critical for injury prevention and rehabilitation.

Football is classified as a sport with one of the highest injury incidence rates among athletes with disabilities; at the London 2012 Paralympic Games, participation in 5-a-side football was deemed the riskiest with 22.4 injuries/1000 athlete-days (*Willick et al., 2013*), while during the Rio 2016 Paralympic Games, the sports with the highest incidence rates were five-a-side football (22.5 injuries/1000 athlete-days), judo (15.5 injuries/1000 athlete-days), and seven-a-side football (15.3 injuries/1000 athlete-days) (*Derman et al., 2018*). Most injuries experienced by amputee football players occur during the game, on synthetic

surfaces, and when interacting with opponents. The authors report that the majority of injuries concerned the lower limbs (*Kasinska & Tasiemski, 2017*). A recent study has shown that amputee football players frequently suffer from play-related injuries. The majority of injuries are located in the lower limbs with the thigh being the most affected area (32.8%), followed by the ankle (13.6%), and knee (9.1%) (*Kasinska, Urbanski & Tasiemski, 2022*).

The ability to maintain and control balance is a complex motor task, and, as such, virtually all neuromusculoskeletal disorders result in some degeneration in this ability. Proprioception is an important element of both static and dynamic balance. Impaired proprioception affects the ability to coordinate movements and maintain balance (*Hansen et al., 2000*). In recent years, computerized posturography has become an essential tool in quantitatively assessing postural stability in clinical settings. The balance systems commonly utilised in sports, assesses an athlete's ability to maintain balance and postural control on multidirectional (*Hosseini et al., 2013*). They are also used in rehabilitation after severe injuries (*Akbari et al., 2015*). Stability tests are used to evaluate athletes and patients with various disorders (*Arifin et al., 2015*; *Zwierko, Lesiakowski & Zwierko, 2020*). Significant losses of playing time and the impact of treatment costs due to lower limb injuries in football underline the need for improved protocols for injury risk reduction. Low balance ability is generally associated with an increased risk of ligament injuries (*Gstottner et al., 2009*; *Rhodes et al., 2000*). Research on handball, football, and tennis confirms the influence of balance training on the occurrence of injuries (*Rhodes et al., 2000*; *Daneshjoo et al., 2022*; *Malliou et al., 2010*). In light of these considerations, the concept and guidelines for optimizing player stability, with a particular emphasis on harnessing the proprioceptive mechanism, gain significance. This approach emphasizes neuromuscular training to enhance the body's proprioceptive feedback loop, leading to better control and coordination during dynamic activities (*Riva et al., 2016*; *Souglis, Travlos & Andronikos, 2023*).

The unexplored domain pertains to investigating the relationship between balance proficiency and sports-related injuries in lower-limb amputees, with a specific focus on the realm of football, where over 80% of physical activities predominantly engage a single lower extremity (*DeLang et al., 2021*). Importantly, the distinguishing attribute of lower-limb amputee players lies in its consistent dependence on the same limb for actions, thereby accentuating the distinct specificity inherent within this domain (*Nowak, Marszalek & Molik, 2022*).

It seems that assessing the level of stability in amputee football players can help evaluate the accompanying risk of sports injuries. Specifically, to our knowledge, the aspects between balance performance and sports injuries have not yet been assessed in lower-limb amputees.

Like other physiological mechanisms, the balance system is susceptible to adaptive modifications resulting from training (*Hughes, Ellefsen & Baar, 2018*). Conducting a solitary assessment at the initiation of the season does not guarantee uniform outcomes at its culmination. Nonetheless, the results obtained from the balance assessment instrument demonstrating stability or minimal fluctuations are acceptable for clinical testing (*Hinman, 2000*), and its test-retest reliability is high when consistent (elevated) resistance levels are employed (*Cug & Wikstrom, 2014*). Investigations aimed at analysing injury occurrences based on diverse pre-season functional or biomechanical parameters predominantly focus

on different time perspectives, *e.g.*, one round/sport season/a few seasons; applying the tests only once (*Watson et al., 2017*; *Šiupšinskas et al., 2019*; *Cichocki & Grygorowicz, 2022*). Following this manner analogous, the present study limited its analysis to the inception stage and refrained from subsequent repetitions.

Therefore, the primary purpose of this study was to examine the usefulness of the balance test in preseason evaluation by calculating between-group differences in stability parameters between injured and non-injured amputee players during one football season.

## MATERIALS & METHODS

### Study design

The study was designed as a pilot prospective observational study. It was based on data collected from a cohort of amputee football players from the Polish National Team.

### Procedures

To meet the criteria of inclusion into the study, the player had to be a member of the Polish National Team and had to actively participate in an official or friendly match during the 2018/2019 season (that is, at least in one international match between September 2018 and June 2019). Every player was tested on postural stability before the 2018/2019 football season. Subsequently, all players were watched for signs of sports injuries sustained during the season. Information about injuries was collected by physiotherapists and doctors of the club and the national team. For the purpose of this study, a sports injury is defined as any injury or trauma that hinders, restricts, or alters one's ability to engage in sports for at least one day (*Bauerfeind et al., 2015*). Written informed consent was obtained from all players invloved in the study, who could withdraw from the study at any moment. The study was conducted in accordance with the Declaration of Helsinki and the Bioethical Commission approved the study at the Medical University in Poznań, Poland (letter dated to February 2, 2017, decision no 138/17).

### Participants

The study involved 92% members of the Polish national amputee football team ($n = 25$), except for goalkeepers with upper limb amputations. The players were selected from the top division of the Polish League and were affiliated with five different clubs. All players in this study were male, with an average age of 29 years (SD = 7.9), a stature of 174.2 cm (SD 5.2) and a body mass of 80.1 kg (SD = 13.1). Almost half (48%) of the players had received a university education. While most participants ($n = 20$) acquired their disability during their lifetime, five individuals reported congenital limb defects. Among those who sustained injuries resulting in amputations, 14 occurred due to traffic accidents, four due to cancer, and two from other diseases. Ten players had undergone above-knee amputations (A2), and ten had undergone below-knee amputations (A4). Six participants identified themselves as strikers, nine as midfielders, and ten as defenders (goalkeepers were excluded from this study; since according to the amputee football rules—goalkeepers must have the same type of amputation as the outfield players, with an upper limb or a lower limb amputation; thus in practice—each goalkeeper has two legs, but only one hand.) On

average, the participants had trained for 3.42 years (SD = 2.1). Approximately 32% of the surveyed athletes had played football before the amputation-inducing incident.

## Data collection

Data were collected as previously described (*Zwierko, Lesiakowski & Zwierko, 2020*). The assessment of static postural control in amputee football players was conducted using the Biodex Balance System SD (Biodex Medical Systems Inc., Shirley, NY, USA). This system evaluates postural stability across 12 dynamic levels, with level 1 representing the least stable and level 12 the most stable (rigid). Levels 1 to 12 correspond to a platform tilt of up to 20 degrees. The evaluation of static postural control included a single-leg stance on a rigid platform (level 12), with all tests conducted while participants maintained open eyes. Subsequently, postural control was measured in three tasks: a single-leg stance with progressively reduced platform stability (levels 8 to 4, randomly selected by the device); a two-leg stance with progressively reduced platform stability (levels 8 to 4); and a single-leg stance with platform stability set at level 4. Athletes performed the tests involving both feet using their everyday prostheses. Each balance task lasted 80 s, consisting of three 20-second trials with a 10-second rest interval in between. Participants were tested barefoot and maintained a forward gaze with their arms crossed over their chests. Before the actual testing, participants completed three 20-second platform stability adaptation trials at levels 12, 8, and 4. We analyzed the overall stability index (OSI), anterior-posterior stability index (APSI), and medial-lateral stability index (MLSI). Typically, scores ranged from 0.0 to 5.0, with higher stability index scores indicating poorer balance control (*Arifin et al., 2015*; *Aydog et al., 2006*). The Biodex Balance System has previously demonstrated high test-retest reliability, with correlation values of R = .94 (OSI), R = .95 (APSI), and R = .93 (MLSI) (*Cachupe et al., 2001*). Validation has also been confirmed (*Söke et al., 2018*; *Rohleder, 2012*). All assessments were carried out on a single day at the Laboratory of Kinesiology, Functional and Structural Human Research Center, University of Szczecin''.

Throughout the season, athletes furnished details about any injury they suffered (the anatomical region, type of injury, the duration of time taken off training while recuperating), and the incident's circumstances (such as the location, the situation on the pitch that led to the injury, the phase of a game or training when the injury happened, and the activity that caused the injury). The club physiotherapist collected data about injuries on the day of the injury. A registry of sports injuries that were consulted with a physician was developed (*Kasinska & Tasiemski, 2017*; *Kasinska, Urbanski & Tasiemski, 2022*). The registry contained details regarding the injury's context, the date when the patient visited the physician, the diagnostic tests that were performed, the diagnosis, the suggested treatment along with its duration, the period during which the patient refrained from training while the injury healed, and the length of any persistent aftereffects of the trauma, if present.

We used an injury definition described by *Ekstrand et al. (2021)*, who define an injury as a musculoskeletal injury that happens while training or competing in a match, which requires reporting the incident to the coach. This type of injury causes discomfort during the activity, necessitates seeking consultation with a physician or physiotherapist, and
prevents the athlete from participating in future training or matches. In the previous study, we provided a more detailed description of the types and characteristics of the injuries analysed (*Kasinska, Urbanski & Tasiemski, 2022*). For this study, we evaluated only lower-leg injuries.

### Statistical analysis

Medians, minimum, maximum and lower and upper quartiles were used for reporting descriptive characteristics and all measured outcomes. We examined the distribution of variables using the Shapiro–Wilk test, and we used the non-parametric Mann–Whitney U test to test for differences between groups of players who had sustained an injury during the tested season (yes) and those who had not (no). The effect size was calculated by the rank biserial correlation, and 95%CI were calculated. The statistical significance for all procedures was set at alpha 0.05. The JASP program version 0.16.4 (Netherlands) was used for all statistical analysis.

## RESULTS

### Descriptive characteristics and number of injuries

In the sample of amputee football players observed in this study ($N = 25$), ten players sustained a lower leg injury during one football season (September 2018–June 2019), and 15 did not sustain an injury in the same season (Table 1). In summary, players suffered quadriceps muscle contusion (three injuries), hamstring, quadriceps, and triceps surae muscle strains (each muscle was injured once), fibular collateral ligament (one player), ankle sprains (two players), and patella dislocation (one injury). We noted no significant group differences in age, weight, height, and time from amputation. However, there was a between-group difference in volume and frequency of training: amputee footballers who sustained an injury trained longer and more frequently during the week than non-injured players (Table 1).

### Between group differences in balance test

Although in most parameters, players from the injured group received lower values than non-injured players (Table 2), no significant between-group differences were observed during the one-leg balance tests in static and dynamic positions. Minimum values for tested outcomes ranged from 0.2 degree to 0.7, while maximum values ranged from 0.5 degree to 4.3. For all analysed indexes, in injured amputee footballers and non-injured players 25% of registered outcomes were lower 0.3 degree and higher then 1.65 degree (Table 2). The between-group differences for analysed outcomes calculated with Mann–Whitney U test did not confirm any significant differences; however, some trends might be visible for OSI ($p = 0.061$) indicating lower median value for OSI one-leg static in injured players (Table 3).

## DISCUSSION

The article aimed to evaluate the usefulness of preseason balance tests as predictors for suffering lower leg musculoskeletal injuries in amputee football players by assessing the

**Table 1 Descriptive statistics for injured (*n* = 10) and non-injured (*n* = 15) players.**

| | Age [yrs] | | Weight [kg] | | Height [cm] | | Time from amputation [yrs] | | Training [yrs] | | Training frequency [times/week] | |
|---|---|---|---|---|---|---|---|---|---|---|---|---|
| | No | Yes | No | Yes | No | Yes | No | Yes | No | Yes | No | Yes |
| Median | 29.00 | 31.00 | 80.85 | 78.95 | 171.00 | 174.00 | 8.00 | 5.00 | 3.00 | 4.75 | 2.00 | 3.00 |
| Std. Deviation | 8.81 | 6.45 | 12.19 | 14.37 | 19.58 | 6.12 | 4.96 | 9.89 | 2.08 | 1.49 | 0.93 | 0.99 |
| Shapiro–Wilk | 0.95 | 0.83 | 0.90 | 0.93 | 0.54 | 0.99 | 0.92 | 0.78 | 0.85 | 0.87 | 0.86 | 0.83 |
| *P*-value of Shapiro–Wilk | 0.59 | 0.04 | 0.10 | 0.46 | <.001 | 1.00 | 0.21 | 0.01 | 0.02 | 0.10 | 0.03 | 0.03 |
| Minimum | 17.00 | 16.00 | 65.60 | 52.70 | 101.00 | 164.00 | 0.00 | 0.00 | 0.50 | 2.00 | 1.00 | 2.00 |
| Maximum | 48.00 | 34.00 | 109.60 | 94.60 | 182.00 | 184.00 | 15.00 | 30.00 | 6.00 | 6.00 | 4.00 | 5.00 |
| 25th percentile | 26.00 | 21.75 | 75.70 | 68.45 | 168.50 | 169.50 | 3.00 | 1.00 | 0.50 | 3.25 | 1.00 | 2.00 |
| 50th percentile | 29.00 | 31.00 | 80.85 | 78.95 | 171.00 | 174.00 | 8.00 | 5.00 | 3.00 | 4.75 | 2.00 | 3.00 |
| 75th percentile | 37.5 | 31.75 | 87.78 | 86.68 | 179.50 | 178.25 | 10.50 | 7.75 | 4.75 | 5.88 | 2.50 | 3.00 |
| Between group differences | *P* = 0.66 | | *P* = 0.44 | | *P* = 0.80 | | *P* = 0.50 | | *P* = 0.03 | | *P* = 0.04 | |

potential differences between injured and non-injured players recorded during one-leg balance tests. We did not prove that preseason values of the overall, anterior-posterior, and medial-lateral stability indexes can reliably predict the risk of lower leg injuries in amputee football.

While considering the potential existence (or absence) of a connection between results from static and dynamic balance assessments and the probability of subsequent sports injuries (along with the broader utility of balance tests as diagnostic tools), it's important to bear in mind that football is categorized as an externally-paced open-skills sport: as such, it abounds in dynamically changing, unpredictable situations in the playing field. Therefore, sports injuries incurred in football may depend on many other factors, some identified in the previous research (*Esmaeili et al., 2018*). On the other hand, it has been observed that amputee football players benefit from their sports training in terms of improvement in their muscular strength, sprint performance, balance, and locomotion (*Ozkan et al., 2012*; *Simim et al., 2017*; *Aytar et al., 2012*) with training facilitating their adaptation to this dynamic game environment. Moreover, the previous findings showed that, compared to able-bodied soccer players, amputee football players showed better adaptation and balance-restoring skills in conditions of increased balance-task difficulty (*Zwierko, Lesiakowski & Zwierko, 2020*). In the case of athletes with disabilities (lower limb amputation), a less visible relationship exists between their balance skills and the risk of falls during locomotion (*Steinberg et al., 2019*). Future longitudinal studies on larger populations of players should confirm or challenge this report's findings, especially on balance tests as predictors of sports injuries in amputee football. The repeated balance measurements (at the beginning of every football season) will gauge changes in this parameter and ascertain whether it improves with years of training. Both two-legged football and amputee football often require their players to make snap decisions: without visual control, relying on changing time and space conditions. Therefore, future research should also investigate players' postural control in scenarios without visual control.

Kasińska et al. (2023), *PeerJ*, DOI 10.7717/peerj.16573

**Table 2** Median, lower and upper quartiles of analysed outcomes for amputee footballers for injured ($n = 10$) and non-injured ($n = 15$) players.

| Tested outcome | Group | Valid | Median | Std. deviation | Shapiro–Wilk | P-value of Shapiro–Wilk | Minimum | Maximum | 25th percentile | 50th percentile | 75th percentile |
|---|---|---|---|---|---|---|---|---|---|---|---|
| OSI one-leg static | No | 15 | 0.70 | 0.52 | 0.83 | 0.01 | 0.50 | 2.30 | 0.60 | 0.70 | 1.20 |
| | Yes | 10 | 0.55 | 0.26 | 0.91 | 0.30 | 0.30 | 1.20 | 0.50 | 0.55 | 0.78 |
| AP one-leg static | No | 15 | 0.50 | 0.46 | 0.71 | <.001 | 0.30 | 2.10 | 0.40 | 0.50 | 0.75 |
| | Yes | 10 | 0.40 | 0.16 | 0.90 | 0.21 | 0.20 | 0.80 | 0.33 | 0.40 | 0.50 |
| ML one-leg static | No | 15 | 0.40 | 0.30 | 0.79 | 0.003 | 0.30 | 1.10 | 0.30 | 0.40 | 0.70 |
| | Yes | 10 | 0.35 | 0.13 | 0.93 | 0.47 | 0.20 | 0.60 | 0.30 | 0.35 | 0.48 |
| OSI one-leg dynamic (8->4) | No | 15 | 1.20 | 0.96 | 0.79 | 0.002 | 0.50 | 4.30 | 1.05 | 1.20 | 1.65 |
| | Yes | 10 | 1.00 | 0.78 | 0.79 | 0.01 | 0.70 | 3.00 | 0.78 | 1.00 | 1.40 |
| AP one-leg dynamic (8->4) | No | 15 | 0.90 | 0.86 | 0.69 | <.001 | 0.30 | 3.90 | 0.80 | 0.90 | 1.15 |
| | Yes | 10 | 0.70 | 0.59 | 0.76 | 0.01 | 0.50 | 2.20 | 0.55 | 0.70 | 0.98 |
| ML one-leg dynamic (8->4) | No | 15 | 0.70 | 0.41 | 0.86 | 0.03 | 0.30 | 1.60 | 0.50 | 0.70 | 0.90 |
| | Yes | 10 | 0.55 | 0.45 | 0.87 | 0.11 | 0.30 | 1.70 | 0.43 | 0.55 | 0.98 |
| OSI SN one-leg dynamic (4) | No | 15 | 1.10 | 0.70 | 0.82 | 0.01 | 0.60 | 3.20 | 1.00 | 1.10 | 1.50 |
| | Yes | 10 | 1.05 | 0.66 | 0.78 | 0.01 | 0.60 | 2.90 | 0.83 | 1.05 | 1.28 |
| AP SN one-leg dynamic (4) | No | 15 | 0.90 | 0.55 | 0.69 | <.001 | 0.50 | 2.80 | 0.70 | 0.90 | 1.05 |
| | Yes | 10 | 0.80 | 0.62 | 0.68 | <.001 | 0.40 | 2.60 | 0.63 | 0.80 | 0.90 |
| ML SN one-leg dynamic (4) | No | 15 | 0.60 | 0.43 | 0.79 | 0.003 | 0.30 | 2.10 | 0.60 | 0.60 | 1.00 |
| | Yes | 10 | 0.70 | 0.22 | 0.91 | 0.29 | 0.40 | 1.00 | 0.53 | 0.70 | 0.85 |

**Table 3** Between-group differences for analysed outcomes calculated with Mann–Whitney $U$ test.

| | W | df | $p$ | Rank-biserial correlation | SE rank-biserial correlation | 95% CI for rank-biserial correlation | |
|---|---|---|---|---|---|---|---|
| | | | | | | Lower | Upper |
| OSI one-leg static | 109.00 | | 0.06 | 0.45 | 0.24 | 0.02 | 0.74 |
| AP one-leg static | 105.50 | | 0.09 | 0.41 | 0.24 | −0.04 | 0.72 |
| ML one-leg static | 101.00 | | 0.15 | 0.35 | 0.24 | −0.11 | 0.68 |
| OSI one-leg dynamic (8->4) | 88.00 | | 0.49 | 0.17 | 0.24 | −0.29 | 0.57 |
| AP one-leg dynamic (8->4) | 91.00 | | 0.39 | 0.21 | 0.24 | −0.25 | 0.60 |
| ML one-leg dynamic (8->4) | 82.00 | | 0.72 | 0.09 | 0.24 | −0.36 | 0.51 |
| OSI SN one-leg dynamic (4) | 89.00 | | 0.45 | 0.19 | 0.24 | −0.28 | 0.58 |
| AP SN one-leg dynamic (4) | 91.00 | | 0.39 | 0.21 | 0.24 | −0.25 | 0.60 |
| ML SN one-leg dynamic (4) | 85.00 | | 0.59 | 0.13 | 0.24 | −0.33 | 0.54 |

**Notes.**

Effect size is given by the rank biserial correlation.

To the best of our knowledge, this is the first study to evaluate the scores of the OSI, APSI, and MLSI regarding lower leg injury in amputee football. Therefore, it isn't easy to compare our results with other studies. However, some previously published studies describe the strategy of applying the balance tests in the amputee players population to investigate the relationship between core stability, balance, and strength in amputee soccer players. The results showed that there was a correlation between flexor isokinetic trunk muscle strength at the velocity of 60°/s and modified plank test and There was a negative correlation between flexor isokinetic trunk muscle strength at the velocity of 180°/s and Oswestry Disability Index score (*Aytar et al., 2012*). Other authors show to evaluate the effect of playing football on balance, strength, and quality of life. Their results show that playing football may have positive effects on balance and HRQOL in patients with unilateral below-knee amputees (*Yazicioglu et al., 2007*).

In a study involving soccer players and utilizing identical balance evaluation parameters, namely OSI, MLSI, and APSI, notable differences were identified between the group of players with chronic ankle instability and the control group, during single-leg standing ($p < 0.05$). The CAI group demonstrated the following values: OSI 7.01 ±2.43 and APSI 5.18 ±1.85, whereas the CTRL group had OSI 5.27 ±1.42 and APSI 3.98 ±1.42. Similar trends were observed during single-leg standing, with the CAI group displaying statistically significant ($p < 0.05$) higher values for the analyzed parameters. The obtained findings reveal that within the group with chronic ankle instability (CAI), platform deflections exceeded those of the control (CTRL) group across all planes. Nevertheless, statistically significant disparities were solely evident for APSI and OSI (*Stefaniak & Bączkowicz, 2018*).

In our study, players who suffered lower leg injuries obtained lower values in most of the tested outcomes related to the postural stability in the preseason evaluation (Table 2). Consequently, the findings of our study may challenge studies that correlate lower OSI values with better results (*Arifin et al., 2015*; *Aydog et al., 2006*). Typically, subjects with injuries or functional problems score higher OSI, MLSI, and APSI values than those without injuries. For example, deficits in ankle arthrokinematic motion and postural control are

present in patients with chronic ankle instability. The CAI group presented higher values of OSI, MLSI, and APSI than the control group (*Bączkowicz, Falkowski & Majorczyk, 2017*). Since, in our study, players with lower leg injuries showed lower values of balance indexes in preseason evaluation, this result was unexpected. However, our results may have been influenced by the fact that the injured players were players with more extended training experience and had a higher frequency of participation in training (Table 1). And while playing more frequently statistically favours a higher number of injuries, the lower OSI score in the more experienced group may also be explained by the fact that playing soccer can positively affect balance (*Yazicioglu et al., 2007*).

Further investigations are needed to understand better this process of balance characteristics in amputee football players. Hence, employing balance tests performed on the Biodex Balance System as an injury risk diagnostic tool requires a further appraisal of this testing system on a larger sample of participants included in the study. It also must be remembered that using any biomechanical and/or functional test for screening players more prone to be injured within the sports season is also questionable; more thorough data is needed in terms of applying different types of evaluation to dichotomously qualify players to the group of higher and lower risk of injuries (*Bahr, 2016*).

The strength of the current article relies on its methodological approach: instead of retrospective data analysis, we observed in real-time as athletes with disabilities incurred sports injuries during one football season. A retrospective study relies on individual recall of former sports injuries: therefore, it may be inaccurate and subject to biases. Our research design made it possible to document all sports injuries incurred in the observation period just after they occurred. All incidents were assessed by an experienced physiotherapist affiliated with the Polish National Team in Amputee Football. Our approach to data collection yielded an accurate, detailed, and comprehensive data set.

On the other hand, the real-time approach limited our analysis to those sports injuries that happened within the 9-month observation period. The relatively small number of sports injuries incurred within that period does not necessarily reflect the usual injury rate across longer intervals, consequently limiting our ability to draw broader conclusions. Another weakness of this study was that the sample consisted solely of players of the National Team in Amputee Football. These well-trained top players know how to act in risky field situations (falls and/or player collisions) to minimize the risk of injuries. Including less experienced players with amputation (perhaps regional level) in future samples could present a more objective picture of the frequency and severity of sports injuries in amputee football. Moreover, although we recruited all members of the national team, the sample is still low, and there is still a disparity in this group between athletes who use prostheses for activities of daily living (ADLs) and those who do not. Thus, it was impossible to analyse the effect of prosthesis on the balance tasks. Therefore, using our results in other groups of athletes should be done critically and with great caution.

## CONCLUSIONS

Our study did not prove that the balance test may be a good predictor of sports injuries in amputee football players. However, from a practical perspective, these results offer a

chance to make a preliminary assessment of fall-related injuries in amputee footballers. Having analysed our sample (collected during the 9-month observation period), we could not confirm that it is possible to predict injuries in amputee footballers concerning their preseason values of the one-leg balance tests.

## ACKNOWLEDGEMENTS

The authors would like to thank all the players who participated in the study.

### Funding
The authors received no funding for this work.

### Competing Interests
The authors declare there are no competing interests.

### Author Contributions
- Zofia Kasińska conceived and designed the experiments, performed the experiments, prepared figures and/or tables, authored or reviewed drafts of the article, and approved the final draft.
- Tomasz Tasiemski conceived and designed the experiments, analyzed the data, authored or reviewed drafts of the article, and approved the final draft.
- Teresa Zwierko analyzed the data, authored or reviewed drafts of the article, and approved the final draft.
- Piotr Lesiakowski performed the experiments, authored or reviewed drafts of the article, and approved the final draft.
- Monika Grygorowicz conceived and designed the experiments, analyzed the data, prepared figures and/or tables, authored or reviewed drafts of the article, and approved the final draft.

### Human Ethics
The following information was supplied relating to ethical approvals (*i.e.*, approving body and any reference numbers):

The Bioethical Committee approved the study at the Medical University in Poznań, Poland (letter dated to February 2, 2017, decision no 138/17).

### Data Availability
The raw measurements are available as Supplemental File.

### Supplemental Information
Supplemental information for this article can be found online at http://dx.doi.org/10.7717/peerj.16573#supplemental-information.

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
