# Peer review of "The usefulness of balance test in preseason evaluation of injuries in amputee football players: a pilot prospective observational study"

_PeerJ, doi:10.7717/peerj.16573_

## Round 0.1 · original submission · Major Revisions

Dear Authors,

The reviewers and I have completed our evaluation of your manuscript and recommend a major revision before re-submission.

Please review the comments and resubmit your revised manuscript.

Reviewer 1 ·

Basic reporting

At the beginning I would like to thank the authors for the opportunity to familiarize themselves with the work on the analysis of the element of balance in combination with the possibilities of predicting injury. In addition, I appreciate the contribution related to the promotion of sports for people with disabilities, and every report of this type made public should influence the awareness of the sports community and confirm the existence of such problems. In connection with the above, I recommend a few corrections which I will try to describe in detail.

The English language used is correct and understandable for the recipient, there are small errors of the nature of the correctness of the sentence, here are some examples.

Line 26 - Correct article usage "the balance"
Line 48 - Change the punctuation "at al.,"
Line 50 - Replace the word - "Despite"
Line 66 - Change the noun form "Sports"
Line 80 - Add the comma
Line 117 - Change preposition
Line 135 - Correct word choice "and"
Line 151 - Add the space and standardize notation of bibliographies
Line 156 - Remove the comma after "and"
Line 171 - Change preposition and change the spelling
Line 211 - Change preposition

In light of the above, I am asking you to reconsider

Paragraph 83 to 96
At this point, I propose to refer to the general idea and indications for work in the field of building stabilization in players. I mean the proprioceptive mechanism here.
In addition, you can look for a correlation with football, in which about 95% of sports activities are dedicated to one leg (literature) of course, the specificity of AMPfutbol is that it is always the same clover.

It is absolutely necessary to remove the product name from the introduction division (the work does not concern the validation of the device)

Experimental design

In the introduction, there is no information confirming the thesis that the result obtained on the machine for assessing the level of the balance does not change or does not change much. Like every mechanism in our body - the balance is subject to adaptive changes related to training. One test at the beginning of the season will not be the same test at the end of the season.
Summing up, all the works trying to predict the occurrence of an injury concern mainly the short term. In this case, the analysis was only at the beginning and was not repeated.
Please refer to this at the beginning of the article.

line 120
This is not a scientific quantification - please change, e.g.: (%). "almost everyone."

line 129
To explain to the reader, it is necessary to briefly justify why goalkeepers were excluded (Rules and two good legs) - it cannot be assumed that all readers will be from the beginning the goalkeeper's task and the specifics of his work in the goal area. The Ampfootball National team is not yet present in all countries in the world.

Validity of the findings

In paragraphs 235 -239
The authors do not provide details and details described in the cited works. Thus, the reader cannot assess what is really an innovative element in relation to other authors - no results.

In paragraphs 257 -258
Is there commercial work commissioned by the manufacturer?
With such a record, the authors suggest the possibility of such an interpretation. Balance diagnostics is used by many manufacturers of other devices, in this case, the paragraph should be reformulated or the issue of stability analysis or in other words, the balance should be generalized.

Additional comments

Due to the subject of sports for people with disabilities and the role it will play in promoting general acceptance, the work should be made public after making a few modifications and filling in the gaps. Best regards and I wish you a lot of success in working with players in AMPFutbol.

Reviewer 2 ·

Basic reporting

Overall well written and conscripted manuscript. There are a number of minor concerns with respect to reporting. The overall structure is sufficient. Please consider the following concerns:

Introduction:
The theoretical framework needs more development. This must include more detail regarding the change in balance via these participants/amputees in general. The more balance change data the better.

Results:
Incomplete data in the paragraph format. There are not enough statistical reports. Consider adding specific p values, 95% CIs, effect sizes, etc. to strengthen the reporting.
Additionally, the paragraphs should also be able to stand alone, like the tables. Do not rely on the table for everything, if you are stating differences between variables, highlight those differences, then reference the tables.

Ln 185. Please include the N descriptors.
Ln 187. Please include a reference to Table 1 here to let the reader know where they can find the descriptors of the groups.

Experimental design

The experimental design was well described and presented however there are a few pieces that need to be addressed:

General: Was there any power analysis performed a priori regarding an appropriate sample size? If so, please include
Ln 110. How were they watched/monitored?
Ln 123. More detailed descriptors are needed here. Height, weight, BF%?
Ln 142-145. Unclear of the specifics of the test here, given the uniqueness of the participants. Given amputee status, who was the 2 leg stance completed? Please provide more detail if you made specific changes to the test.
Ln 149-151. What was the range of scores for these tests? What was the lowest possible vs highest possible score?
Ln 151-152. What about validity? Any evidence to support the validity as well as the reliability you report here? Additionally, what was the reliability score(s)?
Ln 162-165. Any indication of the value/grade of the injury? If so how was it determined? If not, why?

Validity of the findings

Findings seem in line with the question and are presented in a well-organized manner. No major concerns, however, see the design concerns above.

Additional comments

Ln 30 - Include descriptives of subjects (age, height, weight)
Ln 39-40. Consider including the actual P values for the dependent variables.
Ln 48. Remove the bracket and replace it with parentheses
Ln 64-66. You mentioned two distinct models for injuries, but needed to expand on them and how they would fit your project. Please add.
Ln 68. Provide more details on the injury rates and types here.
Ln 72. Cite and provide more specific detail to support this statement.
Ln 76, Provide detailed evidence of risk. Risk scores?

Discussion:
General: While not your primary focus, presenting the number of injuries reported compared to other amputee football athletes' research and able-body athletes' research over the same time period may provide some information regarding risk.
General (ln 233-238): Compare OSI, MLSI, and APSI scores to other amputees or athletes to highlight the uniqueness of the population.

---

## Round 0.2 · Minor Revisions

Dear Authors,

Please resubmit the manuscript as per the final comments from Reviewer 2

Reviewer 1 ·

Basic reporting

The work has been corrected in accordance with the reviewer's suggestions - I have no further comments

Experimental design

The work has been corrected following the reviewer's suggestions - I have no further comments.

Validity of the findings

The work has been corrected by the reviewer's suggestions - I have no further comments.

Additional comments

The work has been corrected per the reviewer's suggestions - I have no further comments.

Reviewer 2 ·

Basic reporting

No comment

Experimental design

No comment

Validity of the findings

No Comment

Additional comments

Reviewing from the track changes document, not PDF. Number lines are different:

General comment:
1. Please review the use of significant digits and be consistent. Typically 2 decimal places is standard practice unless in a p-value that is <0.00 (is 0.001). Your tables are a little distracting because of the extra decimal.

ln 62-89. Very long paragraph. Recommend being more concise and either shrink the paragraph or make 2.
ln 140-142. Is this a separate paragraph? Does not seem to fit. Please address.
ln 160-162. Is this with the previous paragraph?
ln 310. How were you classifying your effect sizes? Were there specific values you were using for the strengths? please be specific.

---

## Round 0.3 · accepted · Accept

Your manuscript has been accepted for publication. Congratulations!